# Synthesis, Structure, and Spectral-Luminescent Properties of Peripherally Fluorinated Mg(II) and Zn(II) Octaphenyltetraazaporphyrins

**DOI:** 10.3390/molecules27238619

**Published:** 2022-12-06

**Authors:** Alexey Rusanov, Natalya Chizhova, Nugzar Mamardashvili

**Affiliations:** 1G.A. Krestov Institute of Solution Chemistry of the Russian Academy of Sciences, Akademicheskaya St. 1, 153045 Ivanovo, Russia; 2Ivanovo State University of Chemistry and Technology, Sheremetevsky Pr. 7, 153460 Ivanovo, Russia

**Keywords:** Mg(II)- and Zn(II)-octa-(2,6-difluorophenyl)tetraazaporphyrins, spectral-luminescent properties, DFT method, geometrical structure, QSPR

## Abstract

The cyclization of di-(2,6-difluorophenyl)maleindinitrile with magnesium(II) and zinc(II) acetates in boiling ethylene glycol is applied to synthesize Mg(II) and Zn(II) complexes of the octa-(2,6-difluorophenyl)tetraazaporphyrin. The compounds are identified by UV–Vis, ^1^H NMR, and mass spectrometry methods. A comparative analysis is performed of the spectral-luminescent properties of magnesium and zinc octaaryltetraazaporphyrinates and their dependence on the number and position of the fluorine atoms in the macrocycle phenyl fragments. The DFT method is used to optimize the geometry of the synthesized complexes. Machine learning methods and QSPR are applied to predict the Soret band wavelength in the UV–V is spectra of the complexes described.

## 1. Introduction

Porphyrazines, or tetraazaporphyrins, possess remarkable and even unique photophysical, photochemical, and electrochemical properties. Porphyrins and their aza-analogues have been recently widely used as photosensitizers in photodynamic therapy and photodynamic diagnostics and as sensors sensitized by solar cell dyes in chemical catalysis and analytical chemistry [1,2,3,4,5,6,7,8,9]. The chemical modification of the peripheral substituents of the macrocycle leads to significant changes in its spectral and electrochemical properties [10,11], making it possible to produce novel polyfunctional materials with desired photophysical properties that can broaden the application range of porphyrazines.

Porphyrins are known to demonstrate their practical efficiency when part of metal complexes. The selective action of metalloporphyrins is the result of the metal atom nature and molecule structure. The specific chemical behavior of substituted metalloporphyrins is mainly caused by the presence of a complex branched macrocycle and strong conjugation of the π–electron system [12]. The method of template cyclotetramerization of porphyrin molecule fragments with a metal or its salt has been used to synthesize ionic and covalent complexes based on porphyrins [13,14,15] and tetraazaporphyrins [16,17,18,19,20]. In this work, cyclization of di-(2,6-difluorophenyl)maleindinitrile with acetates of the respective metals in boiling ethylene glycol was applied to synthesize and identify Mg(II)- and Zn(II)-octa-(2,6-difluorophenyl)tetraazaporphyrin complexes. The choice of the complexes was prompted by the fact that they are quite easily formed through cyclization of the respective salts with nitriles [16,18,19,20] and 3-carboxymethylphthalimidines [13,14] (i.e., they are accessible compounds). Additionally, they are also quite interesting because, being rather labile, these complexes are used in metal exchange reactions to prepare more inaccessible complexes of aza-porphyrins with heavy metals and variable valence metals. Fluorine-substituted metalloporphyrins and their aza-analogs are promising for designing new functional materials with unusual nonlinear optical, catalytic, and electrochemical properties. A comparative analysis was carried out of the spectral-fluorescent properties of magnesium and zinc octaaryltetraazaporphyrinates with different numbers of fluorine atoms in the *ortho* and *para* positions of the macrocycle phenyl fragments.

## 2. Results and Discussion

### 2.1. Synthesis and Spectral-Fluorescence Properties of Mg(II)- and Zn(II)-Octa-(2,6-difluorophenyl)tetraazaporphyrins 

In this work, cyclization of di-(2,6-difluorophenyl)maleindinitrile with magnesium(II) and zinc(II) acetates in boiling ethylene glycol was used to synthesize Mg(II)-octa-(2,6-difluorophenyl)tetraazaporphyrin (**1**) and Zn(II)-octa-(2,6-difluorophenyl)tetraazaporphyrin (**2**) (Figure 1). The synthesis of the Mg(II)- and Zn(II)-octa-(4-fluorophenyl)tetraazaporphyrins was described by us in [19].

It was shown that the interaction of *ortho*-difluorosubstituted diphenylmaleindinitrile with magnesium acetate (in the molar ratio of 1:1) in boiling ethylene glycol for 2 min resulted in the formation of Mg(II)-octa-(2,6-difluorophenyl)tetraazaporphyrin (**1**).

The UV–Vis spectrum of the obtained compound in chloroform had absorption bands with the values of 621, 570, and 365 nm (Figure 1a).

The mass spectrum of the magnesium complex with octa-(2,6-difluorophenyl)tetraazaporphyrin shows the *m/z* 1235.9 signal corresponding to the molecular ion of the obtained compound (Figure 2a).

The ^1^H NMR spectrum of compound **1** in CDCl_3_ shows signals of the *meta* and *para* protons of the phenyl rings at 7.33 and 7.66 ppm (Appendix A).

In a similar way, boiling of di-(2,6-difluorophenyl)maleindinitrile with zinc acetate in ethylene glycol for 1 min led to the formation of Zn(II)-octa-(2,6-difluorophenyl)tetraazaporphyrin (**2**).

Figure 1b shows the UV–Vis spectrum of compound **2** in chloroform with the maxima at 620, 568 and 362 nm. 

The mass spectrum of the zinc complex with octa-(2,6-difluorophenyl)tetraazaporphyrin shows the *m/z* 1275.9 signal corresponding to the molecular ion of compound **2** (the value calculated for C_64_H_24_N_8_F_16_Zn is 1274.4) (Figure 2b). The pattern corresponding to the natural distribution of metal isotopes in the corresponding mass spectra is depicted in the Appendix A. The ^1^H NMR spectrum of compound **2** in CDCl_3_ shows signals of the *meta* and *para* protons of the phenyl rings at 7.36 and 7.68 ppm, respectively (Appendix A). 

The IR spectra of the complexes **1** and **2** are shown in Appendix A. The vibration frequencies of the nonplanar complex **2** (the calculation results are presented in Section 2.2) are shifted to the high-frequency region by 2–3 cm^−1^ compared with those of planar complex **1**.

Table 1 presents the characteristics of the UV–Vis spectra of the synthesized compounds. Aza-substitution is known to cause a blue shift in the Soret band compared with that of the porphyrins without substitution (the pyrrole rings, which are responsible for the Soret band, undergo strong electronic excitation induced by the nitrogen bridge atoms). The UV–Vis spectra of the metal complexes with octaphenylteraazaporphyrins have one high-intensity Q band (I) and a vibrational companion (band II) in the visible region (Table 1).

The fluorescence spectra of the *ortho*-substituted Mg(II)- and Zn(II)-porphyrazines were recorded in tetrahydrofuran at room temperature at the excitation wavelength of λ_max_ = 590 nm (c < 10^−7^mol/L). The standard for compounds **1** and **2** was Zn(II)-octaphenyltetraazaporphyrin with the relative quantum yield of 0.12 in tetrahydrofuran [20]. The fluorescence spectra of standard compound and complexes under investigation are presented in Figure 3.

The spectra were normalized by division by the fluorescence intensity maximum. The quantum yield of fluorescence was calculated by the formula [21].
(1)$$Q_{x}=Q_{st}\frac{I_{x}A_{st}n_{x}^{2}}{I_{st}A_{x}n_{st}^{2}}$$
where *Q_x_* and *Q_st_* are the quantum yields of the sample under study and the standard, respectively; *A_x_* and *A_st_* are their optical densities at the excitation wavelength; *I_x_* and *I_st_* are their respective integral intensities; *n_x_* and *n_st_* are the refraction indices for tetrahydrofuran and the standard (which coincide in our case), respectively. The measurement error was ~10%.

The quantum yield of fluorescence for magnesium complex **1** increases by ~0.11 units compared to zinc complex **2**. The higher quantum yield of the magnesium complex than that of the zinc complex is associated with the atomic weight of the metal constituting the porphyrazine complex. A magnesium atom is known to be a “light” metal, and its complexes in solutions have a higher quantum yield. The quantum yield of fluorescence of the *ortho*-substituted complexes is higher than that of the respective *para*-substituted compounds (Table 2), as the studies showed. This is probably caused by significant changes in the geometry of the *ortho*-substituted Mg- and Zn-porphyrazines in comparison with the respective *para*-substituted compounds. The quantum yield of fluorescence of the perfluorinated Zn(II)-porphyrazine (ZnF_40_PA) in tetrahydrofuran [20] is also 0.02 units higher than that of the *para*-substituted Zn(II)-porphyrazine (ZnF_8_PA) in chloroform (Table 2).

### 2.2. Geometrical Optimization and Energy Characteristics

We carried out geometrical optimization of the Mg(II) and Zn(II) complexes with octa-(2,6-difluorophenyl)tetraazaporphyrin and performed a natural bond orbital (NBO) analysis [22] by the density functional theory (DFT) method [23,24] applying the hybrid Becke three-parameter Lee–Yang–Parr exchange–correlation functional (B3LYP) [25] and the 6-31G++ basis set [26]. The calculations were made in the Gaussian V16 software package.

Figure 4 shows the main designations of the atoms in the structure of the compounds under study.

The geometrical optimization of the studied compounds showed that the most energetically favorable structure was that with C_1_ symmetry. The structure with D_2_d symmetry had imaginary frequencies of the vibrations, the shape of which facilitated the transition to the structure with the C_1_ symmetry. There were four vibrations of that kind. Hence, the D_2_d symmetry was the fourth-order saddle point on the C_1_ path.

For D_2_d symmetry, four imaginary frequencies were found: −27.7561 (A2), −31.8519 (E), −31.8519 (E), and −34.8350 (B1) for MgF_16_PA; and −28.5049 (A2), −32.7111 (E), −32.7111 (E), and −35.7748 (B1) for ZnF_16_PA. The vibrations correspond to a change in the phenyl rings position relative to the macrocycle. The dihedral angle C-C_β_-C_1_^ph^-C_2_^ph^ varies from ~87 to ~90.

At the point of minimum total energy, the matrix of second derivatives has only positive eigenvalues, and, at the saddle point, it has one negative eigenvalue. Because, in our case, there were four negative values, we could consider this a fourth-order saddle point. The total energy minima correspond to stable structures and intermediates, while the saddle points correspond to transition states.

A comparison of the geometric parameters of complexes **1** and **2** showed that the replacement of the metal in the center slightly changed the macrocycle structure but did not affect the geometric parameters of the phenyl rings. The N-Zn bond was shorter and, hence, stronger than N-Mg, which means the zinc coordination interaction with the nitrogen atoms was also stronger.

The phenyl rings were shown to be geometrically different. The dihedral angles in Table 3 demonstrate this. They can be divided into two types for convenience (Figure 5).

The NBO analysis did not give a clear explanation of this structure. The interactions of the orbitals indifferent types of the rings were not the same, but the differences were within the measurement error.

To some extent, the angle N-M-N, where N atoms are not near and are opposite, can serve as a planarity measure. For the Mg complex, this value is 180; this is probably connected with the size of the metal–ion and the ionic type of bond. For the Zn complex, the angle is 172.675, which indicates some nonplanarity degree. The M-N-C-N_α_ angle can also be considered as a planarity measure; for an idle planarity structure, this value would be 0. For compound **1**, this angle is close to zero—0.381; for compound **2**, the value is 4.453.

Using the Mercury 4.2.0 program [Cambridge Crystallographic Data Centre (CCDC), Cambridge, UK], we built a plane using 16 atoms of the macrocyclic core and measured several distances to this “perfectly planar” plane (Appendix A). As we learned by the angles in the structure, the Mg complex has a more planar structure than the Zn complex. On average, the ratio of atom–plane distances in compound **2** to those in compound **1** is about two or three (Appendix A). At a certain point, the atoms of MgF_16_PA are more distant from the “perfectly planar” plane than the atoms of ZnF_16_PA; however, this is balanced by a less-distant neighbor atom to the plane, in contradistinction from ZnF_16_PA.

An analysis was completed of the energy distribution of the molecular orbitals in the vicinity of the highest occupied molecular orbitals (HOMO and HOMO-1) and the lowest unoccupied molecular orbitals (LUMO and LUMO+1), as well as the HOMO-LUMO energy gap widths (∆E) of the magnesium and zinc complexes. Figure 6 presents the type and energy of the frontier molecular orbitals of the studied compounds.

The molecular orbitals involved in the excitation or absorption (e.g., Q band) are shown in Figure 6. The orbital calculation models show that the π–π* character of the transition between HOMO, HOMO-1, LUMO, and LUMO+1 is characteristic of both complexes. This conclusion follows from the localization of the electrons on the π-orbitals of the central part of the macrocycle.

The occupied molecular orbitals of Zn-porphyrazine were lower than those of Mg-porphyrazine, whereas the virtual ones were higher. This explains why the difference between the ZnPz HOMO and LUMO was slightly bigger in absolute value, which means that the π–π* transition was more energy-consuming than that of ZnPz. Quantum chemical calculations showed that the metal replacement (magnesium replacement with zinc) in the macrocycle core had the biggest effect on HOMO-1 as it caused the most significant changes in its energy. Among the frontier orbitals, it was HOMO that was affected most.

We calculated electronic spectra for obtained compounds by the TD-DFT method, using nstate = 10 and root = 1 parameters (Appendix A). We also tried to calculate one spectrum with nstates = 6 and root = 20 parameters (Appendix A). Unfortunately, essential results were not obtained for these parameters. The absorption bands of the Zn complex have a significant red shift in comparison with those of the Mg complex; in the experiment, the Soret band of the Zn complex had little blue shift in comparison with that of the Mg complex. Moreover, the TD-DFT calculation of compound **2** did not predict the active Soret band; it is difficult to say why that is the case, because the structures are rather similar. Perhaps the issue related to a program error or to the calculated electron structure of compound **2** (for example, we can see some differences in virtual orbitals in comparison with complex **1**).

### 2.3. Mathematical Simulation for Prediction of Optical Properties

We previously developed a QSPR mathematical model for the prediction of the absorption wavelength of porphyrins and their analogues [27] on the online portal Online Chemical Database and Modeling Environment (Ochem) [28,29] using five-fold cross-validation [30], the random forest regression (RFR) machine learning method [31] and a consensus model integrating models with several descriptors. We used it to predict the spectral properties of the compounds described in this work, including those from [19,20].

The prediction results did not prove to be promising at this stage (Table 4). Other authors [32] also attempted to predict this value, but their results were as unpromising as ours despite a bigger sampling (Table 4). Such results can easily be explained: the datasets used did not contain any data on either meso-aza-substituted porphyrins or their analogues with a heteroatom in the macrocycle meso-positions. This made the models incapable of accurately predicting the properties of the structures under consideration. However, finding this feature in the obtained mathematical models can facilitate their further development through adding data to the datasets or developing personalized models for tetraazaporphyrins and their derivatives.

Previously, we calculated the electron spectra of obtained compounds using the TD-DFT method, and we compared the results of a machine learning (ML) prediction with the TD-DFT prediction (Table 5).

As can be seen from the comparison of calculated electron spectra characteristics (TD-DFT method) with the results of the machine learning (ML) prediction, the errors are comparable (Table 5). In both cases, there are ways to improve the prognosis. The mathematical model, as indicated, does not quite match the tetraazaporphyrins, because they are not in the training set. A prediction using quantum chemistry was performed by TD-DFT without taking into account the dissolving model, and the solvent has a significant effect on the spectrum band’s position. Both methods have their advantages and disadvantages. However, machine learning methods have recently been extremely dynamically developing, and they are also somewhat easier for a user, because they often do not even need a commercial program.

## 3. Materials and Methods

2,6-Difluoroacetonitrile (Acros, Singapore, Singapore), aluminum oxide (Merck, Rahway, NJ, USA), magnesium and zinc acetates, ethylene glycol (ultra-high purity), and solvents (high purity) were used without additional treatment. The UV–Vis spectra were recorded on a Cary-100 spectrometer at room temperature. The fluorimetric measurements of the metalloporphyrin solutions in tetrahydrofuran (Merck) were carried out on a Shimadzu RF-5301 fluorimeter according to the procedure described in [21].The mass spectra were obtained with a MALDI-TOF Shimadzu Biotech Axima Confidence mass spectrometer (with dihydroxybenzoic acid as the matrix). The ^1^H NMR spectra were registered with a Bruker AV III-500 apparatus (with tetramethylsilane as the internal standard, Singapore, Singapore). IR spectra were recorded on a Fourier spectrometer (VERTEX 80v, Boston, MA, USA). Elemental analyses were performed on a CHN analyzer Flash EA 1112. *Di-(2,6-difluorophenyl)maleindinitrile* was synthesized according to the optimized procedure described in [19]. ^1^H NMR spectrum (CDCl_3_), δ, ppm: 7.62–7.58 m, 7.45–7.41 m (2H, H_Ph_^para^_,_), 7.17, 6.94 dd (4H, H_Ph_^meta^_,_
*J* 7.60) (Appendix A). Mass spectrum, *m/z* (I_rel_., %): 302.45 (98) [M]^+^. The calculated value for C_16_H_6_N_8_F_4_N_2_ is 302.39 (Appendix A).

*Mg(II)-octa-(2,6-difluorophenyl)tetraazaporphyrin* (**1**). A mixture of 0.06 g (0.2 mmol) of di-(2,6-difluorophenyl)maleindinitrile and 0.029 g (0.2 mmol) of Mg(OAc)_2_ in 3 mL of ethylene glycol was heated to the boiling point, boiled in a reflux flask for 2 min, and cooled down. Then, DMF was added. After that, distilled water and NaCl_solv_ were added to the reaction mixture. The precipitate was filtered out, washed with water, and dried. The residue was dissolved in dichloromethane and successively chromatographed on alumina first by dichloromethane, then by chloroform and a chloroform–ethanol mixture (2:1). Yield: 0.018 g (0.0146 mmol, 30%). ^1^H NMRspectrum (d_6_ DMSO), δ, ppm: 7.66 wide s (8H, H_Ph_^para^), 7.33 wide s (16H, H_Ph_^meta^). Mass spectrum, *m/z* (I_rel_., %): 1235.9 (99) [M + 2H]^+^. The calculated value for C_64_H_24_N_8_F_16_Mg is 1233.3. Anal. calcd. for C_64_H_24_N_8_F_16_Mg (%): C 62.33; H 1.96; N 9.09. Found (%): C 62.30; H 1.93; N 9.01. IR spectrum, cm^−1^: ν C-H 2920, 2848; skelet. vibr. 1627, 1592; δ C-H 1011, 987; pyrrole rings 987; γ C-H 762.

*Zn(II)-octa-(2,6-difluorophenyl)tetraazaporphyrin* (**2**). A mixture of 0.06 g (0.2 mmol) of di-(2,6-difluorophenyl)maleindinitrile and 0.037 g (0.2 mmol) of Zn(OAc)_2_ in 3 mL of ethylene glycol was heated to the boiling point, boiled in a reflux flask for 1 min, and cooled down. The processing procedure here was the same as for the first one. Yield: 0.02 g (0.0157 mmol, 32%). ^1^H NMR spectrum (d_6_ DMSO), δ, ppm: 7.68 wide s (8H, H_Ph_^para^), 7.36 wide s (16H, H_Ph_^meta^). Mass spectrum, *m/z* (I_rel._, %): 1275.9 (99) [M + H]^+^.The calculated value for C_64_H_32_N_8_F_16_Zn is 1274.4. Anal. calcd. for C_64_H_24_N_8_F_16_Zn (%): C 60.32; H 1.90; N 8.79. Found (%): C 60.29; H 1.88; N 8.76. IR spectrum, cm^−1^: ν C-H 2922, 2850; skelet. vibr. 1630, 1594; δ C-H 1013, 989; pyrrole rings 989; γ C-H 764.

## 4. Conclusions

In this work, the cyclization of di-(2,6-difluorophenyl)maleindinitrile with magnesium(II) and zinc(II) acetates in boiling ethylene glycol was applied to synthesize Mg(II) and Zn(II) complexes of octa-(2,6-difluorophenyl)tetraazaporphyrin. A comparative analysis was performed of the spectral-luminescent properties of magnesium and zinc octaaryltetraazaporphyrins and their dependence on the number and position of the fluorine atoms in the macrocycle phenyl fragments. The DFT method was used to carry out geometrical optimization of the structure of the synthesized complexes. Machine learning methods and QSPR were employed to predict the Soret band wavelength in the UV–Vis spectra of the complexes described. Although the prediction results did not appear promising at this stage, the present work will certainly facilitate further development of the existing mathematical models by adding data to the datasets or developing models for aza-substituted tetrapyrrole macrocyclic compounds (porphyrins, tetrabenzoporphyrins, and phthalocyanines). Taking into account the fact that fluorine-substituted complexes of porphyrazines have increased n-conductivity, the obtained results can be used to create functional materials exhibiting unusual nonlinear optical, catalytic, and electrochemical properties.

## Data Availability

Not applicable.

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
