# Peer review of "Synthesis, Structure, and Spectral-Luminescent Properties of Peripherally Fluorinated Mg(II) and Zn(II) Octaphenyltetraazaporphyrins"

_molecules, 2022, doi:10.3390/molecules27238619_

Round 1

Reviewer 1 Report

This paper described the successive attempts on the synthesis and theoretical explanations of the photophisical behavior of the metal complexes of porphyrazines.

The paper is of high scientific quality and deserves publication in Molecules in the present form. This opinion is based on the fact that the Authors synthesized two new complexes and fully assigned their structures. The originality of the molecules inder consideration is the involvement of two fluorine atoms in each starting compound thus enabling the formation of octa-aryl substituted organic products containing eight fluorine atoms each. The introduction of fluorine atoms at all ortho-positions leads to the increase of the quantum yields of the photoluminescence of the respective complexes. Quantum chemical calculations supports the experimental findings.

Some little polishing of English should be done, for example: "lately" - "recently" (line 25)

Author Response

Thank you very much for careful reading of our article and for your valuable comments.

Reviewer 2 Report

The paper "Synthesis, Structure, and Spectral-Luminescent Properties of Peripherally Fluorinated Mg(II) and Zn(II) Octaphenyltetraazaporphyrins" is a description of routine (which does not mean, "bad") synthesis of two metal porphyrazines and routine study of their spectra. I feel that this paper would be more suitable for the more specialized journal (for example, Journal of Porphyrins and Phthalocyanines); however, the desicion is up to Editor, of course.

In general, the paper is totally fine. There are just a couple of comments:

1. Why magnesium and zinc were chosen in this paper? Why the peripheral substituents were chosen to be difluorinated phenyl groups? Usually, the justification of choice of objects inder study is given in the Introduction section.

2. Both magnesium and zinc are known to have isotope splitting in mass-spectra as Mg exists as a mixture of 24Mg (79% of isotope abundance), 25Mg (10%) and 26Mg (11%), while zinc consists of three main species 64Zn (49%), 66Zn (28%), 68Zn (18.5%). Could you please demonstrate the pattern corresponding to the natural distribution of metal isotops in your spectra?

3. I have a remark regarding quantum yield (QY) determination and the error of QY claimed to be 10%. Authors use Zn(II)-octaphenyltetraazaporphyrin as a standard and reference the paper [20], where the QY of the latter was determined. It turn, to determine the QY of Zn(II)-octaphenyltetraazaporphyrin, Authors of paper [20] reference zinc complex of phtalocyanine (ZnPc) as a standard, and took QY of the latter from paper [10.1021/ic2027016]. Authors of paper [20] also claim the uncertainty of QY of Zn(II)-octaphenyltetraazaporphyrin to be 10%. In turn, Authors of paper [10.1021/ic2027016]  also determine QY of ZnPc using relative method with an estimated error of 15%, and they reference ZnPc QY taken from the paper [10.1016/0022-2852(69)90335-X] (by the way, the excitation wavelength changes: it was 590 nm in [20], but it was 610 or 618 nm in [10.1021/ic2027016]). Finally, paper [10.1016/0022-2852(69)90335-X] reference the paper [10.1039/TF9575300646] (where the error of QY is extimated to be 7 to 10%), where the absolute QY was determined, and the chain of references is terminated at last. I believe, the errors along this chain of papers should stack somehow, shouldn't they? Could you clarify, please, how exactly the error of 10% was estimated in your paper?

4. Table 2 should include the errors of QYs determined.

5. Would you kindly provide NMR and IR spectra of the studied metal complexes in Supplementary File?

6. The UV-Vis spectra of your compounds can be calculated using quantum chemistry methods (TD DFT) and compared with the experimental data. These calculations can probably be more accurate than machine learning prognosis.

7. What is the possible application of your compounds? The perspective can be added to the Conclusion section.

Author Response

(The authors gave the same response as above.)

Author Response

(The authors gave the same response as above.)

Reviewer 4 Report

The paper by Mamardashvili and coworkers describes the synthesis, structural characterization, and optical properties of peripherally fluorinated Mg(II) and Zn(II) tetraazaporphyrin (porphyrazine, PA) derivatives. The authors previously reported the octa-(4-fluorophenyl)-substituted porphyrazine (MF8PA, RJOC 2019, ref 19). To get insight into the structure and (optical) properties relationship, they synthesized more fluorinated derivatives (MF16PA) starting from di-(2,6-difluorophenyl)maleindinitrile in ca. 30% yields. The absorption and fluorescence spectra were measured and compared with other derivatives. With the DFT method, optimized structures and electronic structures of frontier orbitals were elucidated. Finally, the mathematical simulation using machine learning methods was attempted to predict the optical properties, but the results showed significant errors.  

This reviewer thinks the results presented in this manuscript are very specific and would be attractive only in the porphyrin community. In addition, the presentation styles could be better organized: 4 figures for adoption spectra and mass charts are better to put together, including emission spectra; energy diagrams are better shown together with Table 4; TD-DFT calculations are required to discuss the absorption energy). Thus, the manuscript cannot be accepted without major revision. 

Minor points:

  1. Ref numbers (16–18) are missing or not shown sequentially.
  2. Line 132: D2D => D2d
  3. Please check the references carefully. 

Ref 1: De

Ref 2: e201960077

Ref 9: 511-515

Ref 20: .Lakowicz

etc.

Author Response

(The authors gave the same response as above.)

Round 2

Reviewer 2 Report

Authors made necessary improvements to their manuscript and it looks much better in my opinion. There are only a couple of minor comment left:

1. Please, provide in ESI the full 1H NMR spectra of the synthesized compounds. The spectral fragments provided in the current version can be kept as insets.

2. How many states of interest and roots did you consider during TD DFT calculations? It seems to me that the default number of six was set into calculations, which is actually not enough. Due to some reason, the calculation of electron transitions begins with the most longwave peak; therefore, you might have missed the most of short-wavelength transitions including the Soret band. I would recommend resolving no less than 20 states (it would look like td=(nstates=6,root=20) in Gaussian input file.)

Author Response

Thanks for your valuable comments. We are ready to provide additional information if necessary. The authors.

Reviewer 3 Report

The revised version of the manuscript is greatly improved. There are two points that need a little more.

The authors corrected the few poor word choices or phrasings that were specifically mentioned in the original review but there are a great many more (too numerous to list all) that need to be addresses as was indicated in the original review.  Closer attention to this will greatly improve the manuscript.

On lines 184-186 the authors indicated that they used Hg to provide an estimate of the planarity of the ring. This is a good improvement and is noted in the Supplemental. Can the equations for the two planes be added?

Author Response

(The authors gave the same response as above.)

Reviewer 4 Report

I'm satisfied with the authors' revision.

Minor comments:

1. The citation of ref 2 is not appropriate. "1-9" means a number of pages, not real page numbers. Please use manuscript ID (e201960077).

2. The meaning of "t0" on the arrow in the Scheme is unclear. Please modify it.

Author Response

(The authors gave the same response as above.)
